# Arrhythmogenic Cardiomyopathy—Further Insight into the Clinical Spectrum of Desmoplakin Disease

**Joanne Simpson [1,***, Joan Anusas [1], Denise Oxnard [2], Sylvia Wright [3], Ruth McGowan [2] and Caroline Coats [1]**

1   Cardiology Department, Inherited Cardiac Conditions Clinic, Queen Elizabeth University Hospital,
    1345 Govan Road, Glasgow G51 4TF, UK; joan.anusas@ggc.scot.nhs.uk (J.A.);
    caroline.coats@ggc.scot.nhs.uk (C.C.)
2   West of Scotland Centre for Genomic Medicine, Queen Elizabeth University Hospital,
    1345 Govan Road, Glasgow G51 4TF, UK; denise.oxnard@ggc.scot.nhs.uk (D.O.);
    ruth.mcgowan@ggc.scot.nhs.uk (R.M.)
3   Department of Pathology, Queen Elizabeth University Hospital, 1345 Govan Road, Glasgow G51 4TF, UK;
    sylvia.wright@ggc.scot.nhs.uk
*   Correspondence: joanne.simpson@ggc.scot.nhs.uk

**Abstract:** Arrhythmogenic cardiomyopathy is a familial heart muscle disease characterized by structural, electrical, and pathological abnormalities. Recognition of left ventricular (LV) involvement in arrhythmogenic right ventricular cardiomyopathy (ARVC) has led to the newer term of arrhythmogenic cardiomyopathy (ACM). We report on a family with autosomal dominant desmoplakin (DSP) related ACM to illustrate the broad clinical spectrum of disease. The importance of evaluation of relatives with cardiac magnetic resonance imaging and consideration of genetic testing in the absence of Task Force diagnostic criteria is discussed. The practical and ethical issues of access to the Guthrie collection for deoxyribonucleic acid (DNA) testing are considered.

**Keywords:** arrhythmogenic cardiomyopathy; desmoplakin; blood spot; Task Force criteria; arrhythmogenic right ventricular cardiomyopathy

## 1. Background

The proband died aged 19 years whilst trekking in India. After two autopsies the cause of death was determined as sudden arrhythmic death syndrome (SADS). *"There was no blood or tissue specimen available to be used as source of deoxyribonucleic acid (DNA) for comprehensive genetic analysis in the proband. We obtained access to a historic DNA sample retained from the neonatal Guthrie screening card; this allowed targeted DNA analysis when a variant was identified in another relative."* The pedigree is shown in Figure 1. The family was referred for screening.

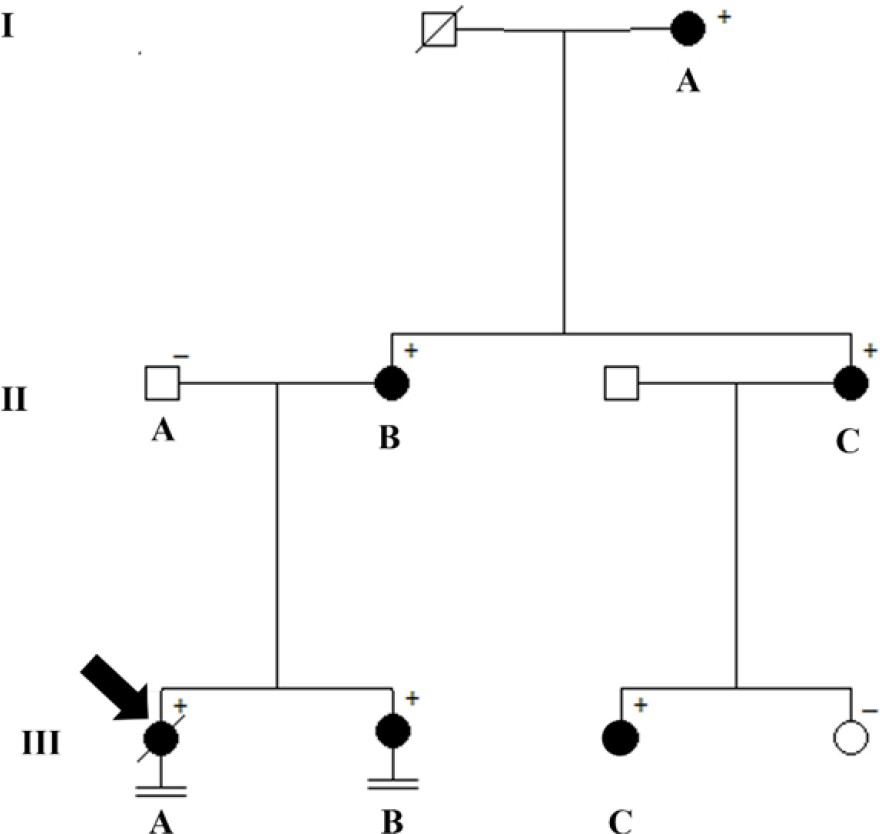

**Figure 1.** Family pedigree. Squares and circles indicate male and female family members, respectively. Symbols with a single slash mark are deceased family members. Arrows indicate proband. Open symbols indicate unaffected family members, and symbols with a cross indicate family members who did not have a clinical evaluation. The presence (+) or absence (−) of a desmoplakin (DSP) variant is indicated for family members with DNA samples available for testing.

## 2. Family Evaluation

A summary of the results of investigations in family members is shown in Table 1.

III:B Sister of proband, an athlete. Initial investigations included 12 lead electrocardiograph (ECG), exercise tolerance test (ETT), and 2-D transthoracic echocardiogram, all of which were normal. She presented six months after the initial evaluation with chest pain after a flu-like illness. During this hospital admission, high-sensitivity Troponin I (hsTnI) measured greater than 5000 pg/nl and non-sustained ventricular tachycardia was recorded. Cardiac magnetic resonance imaging (MRI) showed mid-wall late gadolinium enhancement in the left ventricular inferior septum. No abnormalities of right ventricular size, function, or tissue characterization were detected. Twelve lead ECG showed interventricular conduction delay, prominent U waves, and loss of QRS voltage in the limb leads, when compared to the index evaluation. Holter monitoring showed 663 ventricular ectopics in a 24 h period, which were multifocal in nature. Repeat cardiac MRI 18 months later showed global myocardial edema and left and right ventricular epicardial late gadolinium enhancement in addition to the previously observed mid-myocardial enhancement (Table 1). Endomyocardial biopsy was performed to exclude an alternative cause for the presentation with presumed myocarditis. Microscopy showed a focal increase in intramyocardial interstitial collagen with small foci of replacement fibrosis (Figure 2). A varying degree of myocyte hypertrophy with patchy vacuolar change and myocytolysis was observed. There was no evidence of active myocarditis at multiple histological levels. Overall, the histological features were compatible with a cardiomyopathy that is difficult to classify on the basis of this biopsy alone. The case was reviewed by an expert cardiovascular pathologist in a tertiary referral centre and the possibility of arrhythmogenic

cardiomyopathy (ACM) was raised, although it is recognised that the histological features did not meet Task Force criteria.

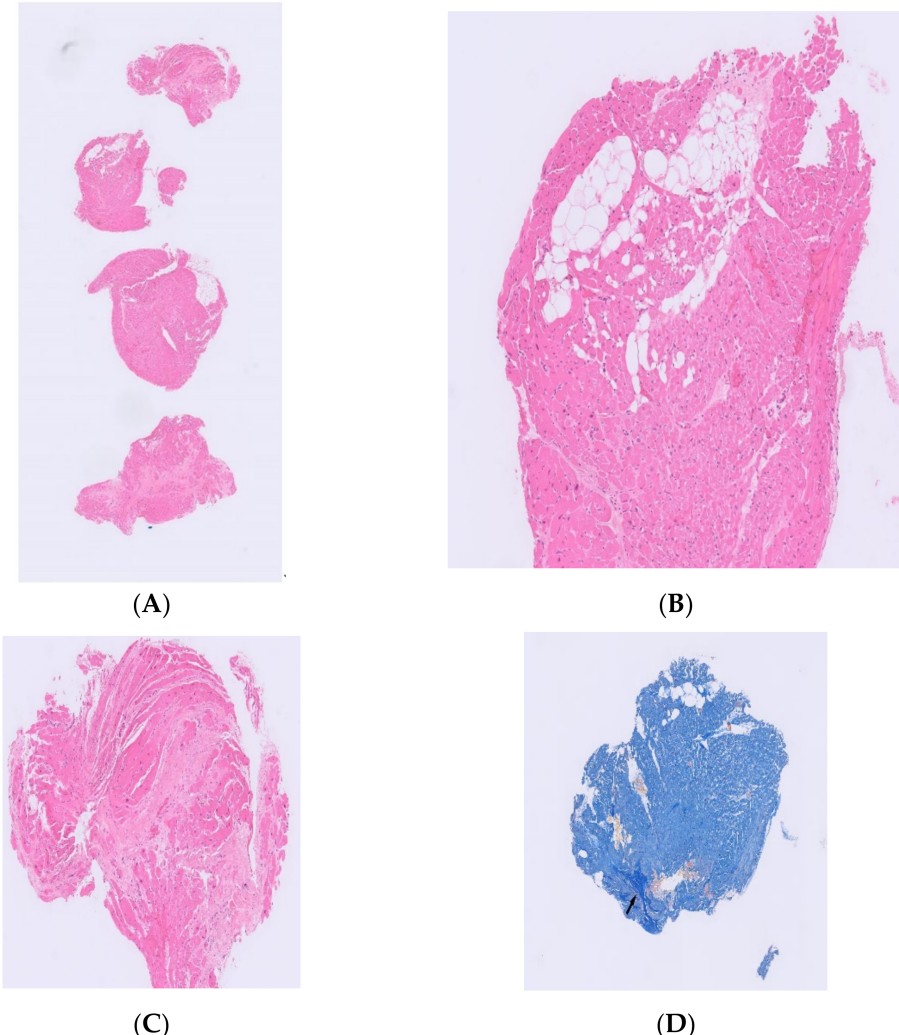

|  |  |
|:---:|:---:|
| (**A**) | (**B**) |
| (**C**) | (**D**) |

**Figure 2.** (**A**) Ultra low power image (H&E). In this image, four fragments of endomyocardium are included for assessment. Note the presence of adipose tissue in some of the fragments. Other fragments appear to show some fibrous replacement of the cardiac myocytes. (**B**) Medium power (×10 H&E). This medium power image shows myocardium with areas of interstitial fibrosis. There is mature adipose tissue and a suggestion of associated fibrous tissue at the top of the section, but this was not demonstrated on tinctorial staining. There is myocyte nuclear hypertrophy and focal areas of myocytolysis were seen. No increase in the cellularity of the interstitium was seen. No granulomas were identified. (**C**) This medium power (×10, H&E) image of another fragment from the biopsy set shows myocardium with areas of replacement fibrosis. Tinctorial staining for excess iron and amyloid deposition were negative (not shown). (**D**) In this medium power image (×10 MSB), the arrow demonstrates an area of fibrosis. No definite established fibrosis can be seen in association with the adipose tissue in this section.

Given the evolution of clinical findings in this family member and the history of sudden arrhythmic death (although not confirmed ACM or ARVC/D) in a first degree relative, a diagnosis of ACM was suspected. According to the revised Task Force Criteria for the Diagnosis of ARVC/D, the non-sustained ventricular arrhythmia (arrhythmia-1 minor) and the sudden death of a family member under the age of 35 years with suspected ARVC/D (family history-1 minor) suggest a possible diagnosis of ARVC/D. After genetic

testing, a shared decision was made between the clinical team and patient for implantation of a subcutaneous implantable cardioverter defibrillator.

II:B Mother of proband. The mother was asymptomatic at the initial clinical evaluation and the 12 lead ECG and echocardiogram were normal by Task Force criteria. The ECG showed low voltage QRS complexes in the limb leads. Cardiac MRI revealed septal mid-wall and inferolateral epicardial late gadolinium enhancement (Table 1). Prior to genetic evaluation of the family, II:B met one minor criteria for a diagnosis of ARVC/D by the revised Task Force criterion.

II: C. Maternal aunt. A normal 12 lead ECG and echocardiogram were recorded at the initial evaluation. Cardiac MRI also showed mid-wall septal late gadolinium enhancement (Table 1).

III:C Maternal cousin. At the time of the initial evaluation, the proband's cousin was asymptomatic. A normal 12 lead ECG and echocardiogram were recorded. Cardiac MRI also showed septal mid-wall late gadolinium enhancement (Table 1).

I: A. The proband's maternal grandmother had New York Heart Association (NYHA) functional class II heart failure and palpitations at the index evaluation. She had an abnormal 12 lead ECG with interventricular conduction delay in the inferior leads with T wave inversion extending from V1-V6 in the precordial chest leads. The echocardiogram showed left ventricular systolic dysfunction. Cardiac MRI showed fibrofatty infiltration of the right ventricle with epicardial and mid wall late gadolinium enhancement in the basal inferolateral wall.

Familial evaluation with the above investigations resulted in the identification of four further individuals as being affected by ACM. Of the individuals evaluated, only one fulfilled the Task Force criteria for a definite diagnosis of ACM or ARVC/D. The revised Task Force criteria for a diagnosis of ARVC/D for each family member are shown in Table 1 [1].

**Table 1.** Summary of investigations and the revised Task Force criteria.

| | Age | Sex | Clinical Presentation | 12 Lead ECG | Transthoracic Echo | Holter | CMR | Histology | Revised Task Force Criteria [1] | Padua Criteria |
|---|---|---|---|---|---|---|---|---|---|---|
| III:A | 19 | F | Sudden cardiac death | X | X | X | X | X | X | X |
| III:B | 17 | F | Palpitations and chest pain |  | LVIDd 4.56 cm LVEF 69% RV minor 3.0 cm Normal RV size (RVOT PLAX 28 mm) and function with no regional akinesia/dyskinesia. | 663 ventricular extrasystoles with LBBB morphology/24 h. Trigeminy and ventricular couplets. |  LVEF 61% LVEDV 149 mL LVEDV/BSA 80 mL/m$^2$ RVEF 52% RVEDV 155 mL RVEDV/BSA 87 mL/m$^2$ Mid apical and anteroseptal myocardial oedema. Global subepicardial and midwall LGE. Elevated T1 1124 ms and T2 79 ms. | Focal increase in intramyocardial interstitial collagen with small foci of replacement fibrosis. A varying degree of myocyte hypertrophy with patchy vacuolar change and myocytolysis. | 2 minor criteria Possible diagnosis | RV: minor (family history) LV: 1 major (Structural myocardial—CMR) and 1 minor (Depolarization abnormalities-low QRS voltage) Dominant left ACM |

**Table 1.** *Cont.*

| | Age | Sex | Clinical Presentation | 12 Lead ECG | Transthoracic Echo | Holter | CMR | Histology | Revised Task Force Criteria [1] | Padua Criteria |
|---|---|---|---|---|---|---|---|---|---|---|
| II:B | 51 | F | Asymptomatic |  Low voltage QRS | LVIDd 5.0 cm LVEF 60% RV normal size No regional akinesia/dyskinesia. | R and L PVCs 81 ventricular extrasystoles/24 h |  LVEF 59% LVEDV 117 mL LVEDV/BSA 65 mL/m$^2$ RVEF 72% RVEDV 91 mL RVEDV/BSA 51 mL/m$^2$ Subepicardial and mid-wall LGE in basal to mid myocardial segments. | Not available | 1 minor * (Family history) | RV: 1 minor (family history) LV: 1 major (structural myocardial) and 1 minor (depolarization abnormalities) Dominant left ACM |
| II:A | 53 | M | Asymptomatic | Not available | LVIDd 4.3 cm LVEF 63% Normal RV size and function with no regional akinesia/dyskinesia. | Not available |  LVEF 63% LVEDV 146 mL LVEDV/BSA 71 mL/m$^2$ RVEF 64% RVEDV 141 RVEDV/BSA 72 mL/m$^2$ No RWMA No LGE | Not available | 1 minor * (Family history) | RV: 1 minor (family history) LV: Nil |

**Table 1.** *Cont.*

| | Age | Sex | Clinical Presentation | 12 Lead ECG | Transthoracic Echo | Holter | CMR | Histology | Revised Task Force Criteria [1] | Padua Criteria |
|---|---|---|---|---|---|---|---|---|---|---|
| I:A | 71 | F | Heart failure—NYHA functional class II. Palpitations. | T wave inversion V1-V6 Interventricular conduction delay in inferior leads | LVIDd 5.33 cm LVIDd index 3.23 cm/m$^2$ Increased RV wall thickness (6 mm) Mild RVSD (TAPSE 15 mm) Abnormal RV apex with localised dilatation and regional wall motion abnormality. | 5280 ventricular extrasystoles/24 h (4.8% total) R and L PVC Bigeminy and trigeminy | LVEDV Normal (No quantification available) RVEDV Normal (No quantification available) RV hypertrophy and fatty infiltration. Focal thickening of lower right ventricular wall. Epicardial LV LGE and mid myocardial LV LGE. | Not available | 1 major and 3 minor from different categories * Definite diagnosis | RV: 3 major (1 morpho-functional ventricular abnormality, 1 repolarization abnormality, 1 ventricular arrhythmia) LV: 1 major (1 structural myocardial abnormality) and 2 minor (repolarization abnormality and depolarization abnormality) |
| II:C | 49 | F | Occasional palpitations | | LVIDd 4.08 cm LVIDd index 2.35 cm/m$^2$ LVEF 60% Normal RV cavity size and function with no regional akinesia/dyskinesia RV minor 2.7 cm | 5 ventricular extrasystoles /72 h | LVEF 76% LVEDV 128 mL LV EDV/BSA 73 mL/m$^2$ RVEF 59% RVEDV 125 mL RVEDV/BSA 71 mL/m$^2$ Subtle mid-wall LGE in mid septum | Not available | 1 minor * (Family history) | RV: 1 minor (family history) LV: 1 major (structural myocardial) |

**Table 1.** *Cont.*

| | Age | Sex | Clinical Presentation | 12 Lead ECG | Transthoracic Echo | Holter | CMR | Histology | Revised Task Force Criteria [1] | Padua Criteria |
|---|---|---|---|---|---|---|---|---|---|---|
| III:C | 22 | F | Asymptomatic | | LVIDd 4.6 cm LVIDd index 2.5 cm/m$^2$ Normal RV size and function with no RWMA | No ventricular extrasystoles | LVEF 61% LVEDV 142 mL LVEDV/BSA 74 mL/m$^2$ RVEF 60% RVEDV 148 mL RVEDV/BSA 77 mL/m$^2$ No RWMA Minor linear LGE in basal inferoseptum | Not available | 1 minor * (Family history) | RV: 1 minor (family history) LV: 1 major (structural myocardial) |

BSA, body surface area; CMR, cardiac magnetic resonance; ECG, electrocardiograph; F, female; LGE, late gadolinium enhancement; LV, left ventricle; LVEF, left ventricular ejection fraction; LVIDd, left ventricular internal diastolic diameter; LVEDV, left ventricular end diastolic volume; LBBB, left bundle branch block; M, male; NYHA, New York Hear Association; PLAX, parasternal long axis view; RV, right ventricle; RVEF, right ventricular ejection fraction; RVEDV, right ventricular end diastolic volume; RVOT, right ventricular outflow tract; RWMA, regional wall motion abnormality; RVSD, right ventricular systolic dysfunction; ACM, arrhythmogenic cardiomyopathy; PVC, right (R) and left (L) premature ventricular contractions. * Prior to genetic evaluation in patient III:B.

### 3. Genetic Testing

Despite not meeting the Task Force criteria for a diagnosis of ACM, after detailed clinical evaluation, there was a high clinical suspicion of ACM in III:B. Next generation sequencing was undertaken for an ACM (nine gene) panel identifying two sequence variants. The variants were classified according to the American College of Medical Genetic classification [2]. A heterozygous desmoplakin (DSP) variant (c.1288G>T; p.(Glu430*)) was predicted to result in the formation of a premature translational stop signal, resulting in an absent or disrupted protein product and therefore was considered pathogenic. *'The DSP variant is predicted to undergo nonsense-mediated decay and therefore PVS1 was applied, along with PM2 as it is not recorded on gnomAD.'* The variant is not present in population databases (no frequency on the Exome aggregation Consortium, ExAC) and has previously been described in individuals with ARVC [3,4].

In addition, a desmocollin-2 (DSC2) variant was identified (c.2335G>T; p.(Gly779Arg), which was considered to be a variant of uncertain clinical significance. The missense variant was predicted to result in the substitution of a glycine by an arginine residue at codon 779, which is conserved through evolution and in silico analysis indicated that the substitution may not be tolerated by the protein. This variant has been recorded in very low frequency (0.009%) on the ExAC database and reported previously as a variant of uncertain clinical significance on ClinVar (NCBI, Bethesda, MD, USA).

Following the identification of the genetic findings in patient III:B, DNA was obtained from the Guthrie card of the proband (III:A). Targeted Sanger DNA sequencing confirmed the presence of the pathogenic variant in DSP but not the variant of unknown significance in DSC2. Cascade predictive genetic testing was performed, looking for the presence of the pathogenic variant in DSP, and was found in the proband's mother (II:B), maternal grandmother (I:A), maternal aunt (II:C) and maternal cousin (III:C).

### 4. Discussion

In this case example, the proband's sister presented after reassuring clinical screening with chest pain, elevated hsTnI, and non-sustained ventricular arrhythmia. The clinical evolution and history of sudden death in the family mounted sufficient clinical suspicion to initiate genetic evaluation, despite not meeting the Task Force criteria for a diagnosis of ACM. This allowed cascade predictive genetic testing in family members for the detection of gene carriers and appropriate evaluation of risk and clinical follow-up. In this case, clinical evaluation of family members with ECG, holter monitoring, and echocardiogram may not have detected a phenotype in three out of four individuals. The clinical diagnosis of ACM in relatives is often difficult because of the low penetrance of mutations and variable expressivity. The proband in this case example presented in the second decade, illustrating the age-dependent penetrance of ACM [5,6].

The presentation of patient III:B with acute chest pain and elevated troponin levels is in keeping with the previously described 'hot-phase' of the condition. In a recent retrospective analysis of 23 patients with ACM meeting the clinical definition of a 'hot-phase' nearly 40% of patients were diagnosed with ACM at the time of their first presentation. In 60% of the patients, an alternative diagnosis was made at the index presentation (myocarditis, acute myocardial infarction, acute pericarditis, and unstable angina). This highlights the importance of distinguishing between acute myocarditis and the hot phase of ACM and reinforces the importance of a detailed family history in such patients.

Historical nomenclature and diagnostic criteria do not account for left ventricular involvement in this condition or for cardiac MRI tissue characterisation findings [7]. It is increasingly recognised that patients with DSP-associated cardiomyopathy frequently have left ventricular involvement and that the subepicardial pattern of late gadolinium enhancement is common [8,9]. This may account for the poor sensitivity of the revised Task Force criteria when diagnosing patients with ACM associated with desmoplakin. The recently published Padua criteria propose the inclusion of left ventricular late gadolinium enhance-

ment and low QRS voltage to diagnose the previously overlooked LV phenotype [10,11]. The Padua criteria are shown in Table 1.

In this case, use of the stored blood spot (Guthrie) card to obtain DNA from the deceased for testing for the variants found in the affected sister provided insight and understanding into the otherwise unexplained death and important answers for the family. This blood spot test obtained on day five of a newborn's life is used to screen for serious health conditions that can be reversed with early treatment, for example, phenylketonuria. The cause of sudden cardiac death in a young person can be challenging to identify. Often a drawn blood sample or tissue storage at post-mortem are not available for the deceased. The dried blood spot provides the opportunity for DNA extraction and advances in genomic testing technology allow genetic testing and confirmation of a familial variant when an alternative stored sample is otherwise unavailable. A recent study of 22 patients with a post-mortem diagnosis of ARVC found clinically relevant variants in 63% of families on whole exome sequencing of dried blood spots [12]. Of the variants detected, four were in ARVC-associated genes and a further six were associated with arrhythmia genes. This has important implications for preventing sudden cardiac death in surviving family members. In Scotland, the Guthrie collection system began in 1965 and there are now over 3 million samples stored. Only from 2003 were parents asked for consent for storage of their children's samples. Whilst in the presented case, the Guthrie card was relevant for an individual and their family, obviously, this unique dataset raises potential opportunities for genetic-based and wider health research. It is also particularly relevant for families with a historical diagnosis of cardiomyopathy where genetic testing did not take place or in those affected by sudden cardiac death. Navigating the future use of the Guthrie cards is currently under consideration by the Scottish Government and a document published in 2014 exists to determine the legal status, the role of consent and governance, and importance of transparency in processes involving the collection [13]. Public engagement with this process has highlighted important areas for future consideration including the possibility that information obtained from the Guthrie card could disadvantage individuals, or the potential for the neonatal screening panel to be extended to screen for variations causing a wide variety of diseases.

## 5. Conclusions

Individuals with DSP-related ACM display a range of clinical phenotypes and thorough cardiac investigation should be undertaken, especially where there is a family history of sudden cardiac death. Advances in genomic testing technology allows the confirmation of genetic variants on stored blood spot samples and in the case of sudden cardiac death provides confirmation of the diagnosis and closure for the family.

**Author Contributions:** Data acquisition, J.S., J.A., D.O. and S.W.; processing and interpretation, J.S., S.W. and C.C.; writing original draft preparation, J.S.; clinical evaluation, J.A., D.O., R.M. and C.C.; study concept, R.M. and C.C.; critical revision of manuscript, R.M.; data curation, C.C.; supervision, C.C. All authors have read and agreed to the published version of the manuscript.

**Funding:** This research received no external funding.

**Institutional Review Board Statement:** Not applicable.

**Informed Consent Statement:** Informed consent was obtained from all subjects involved in the study.

**Data Availability Statement:** The data that support the findings of this study are available from corresponding authors upon request.

**Conflicts of Interest:** The authors declare no conflict of interest.

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
