# Peer review of "Arrhythmogenic Cardiomyopathy—Further Insight into the Clinical Spectrum of Desmoplakin Disease"

_cardiogenetics, doi:10.3390/cardiogenetics11040022_

Round 1
Reviewer 1 Report
The manuscript is well written and provides a comprehensive clinical evaluation of the proband and his family, describing the variability in disease presentation and the fact that the revised Task Force diagnostic criteria for ARVC/D although useful for determining a clinical diagnosis, may not capture the breath of the spectrum of clinical variability in families with ACM.
The authors stated: “There was 24 no deoxyribonucleic acid (DNA) available for genetic testing, nor tissue obtained for further analyses ”. However, they later added that a Guthrie card become available for targeted Sanger sequencing. Please reword the text to clarify your statement such as: “Unfortunately, there was no sufficient blood or tissue specimen to be used as source of DNA for a comprehensive genetic analysis in the proband. We had access to a Guthrie card, which could only allow for targeted DNA analysis.” Please feel free to use this suggestion or clarify in a different manner.
Although, given the high suspicion about ACM in the family, the analysis of a small gene panel (the authors stated that 9 genes were tested) may seem appropriate, given that they did not fulfil the Task Force diagnostic criteria for ARVC/D, a comprehensive cardiomyopathy and arrhythmias gene panel should have been selected. Please provide details about the genes tested (as Supplemental info will suffice) and if copy number variants (CNV) in addition to single nucleotide variants (SNV) were evaluated in the assay.
Given the availability of the gnomAD v2.1.1 ad gnomAD v3.1 database, I suggest to provide the minor allele frequency (MAF) of the DSP variant using the gnomAD database.
Reviewer 2 Report
This is an interesting report of a family with DSP-gene mutation and left-dominant ACM variant. The case highlight several teaching points:
1) the lack of sensitivity of the 2010 ITF for left-dominant ACM variants
2) the possible presentation with a "hot-phase" (sister of the proband)
3) the variable phenotypic expression among family members (from no structural abnormalities, to classic ARVC in the grandmother, to left-dominant phenotypes)
4) the possible use of the Guthrie card for post-mortem genetic testing
5) the importance of Low QRS voltages as a LV involvement marker (J Am Heart Assoc. 2018 Nov 20;7(22):e009855.)
Overall, this was a very interesting case. I have only some minor suggestions:
1) case IIIB: two MRI scans were performed with very different results: this is strange and more details should be provided (was it an error in reporting? What was the time interval between the two scans?).
2) I would discuss briefly the "hot phase" presentation of the disease (Europace. 2021 Jun 7;23(6):907-917. doi: 10.1093/europace/euaa343.)
3) In the table I would add a column with the Padua criteria, that were specifically design to increase the sensitivity for left dominant variants
4) In the table the grandmother is labeled case IIIA, but it should be IA
